# Color Point Cloud Registration Algorithm Based on Hue

Siyu Ren ⬤, Xiaodong Chen *, Huaiyu Cai, Yi Wang ⬤, Haitao Liang and Haotian Li ⬤

Key Laboratory of Opto-Electronics Information Technology, School of Precision Instrument Opto-Electronics
Engineering, Tianjin University, Ministry of Education, Tianjin 300072, China; rsy6318@tju.edu.cn (S.R.);
hycai@tju.edu.cn (H.C.); koala_wy@tju.edu.cn (Y.W.); htliang@tju.edu.cn (H.L.); lihaotian@tju.edu.cn (H.L.)
* Correspondence: xdchen@tju.edu.cn; Tel.: +86-22-2740-4535

**Abstract:** ICP is a well-known method for point cloud registration but it only uses geometric information to do this, which will result in bad results in some similar structures. Adding color information when registering will improve the performance. However, color information of point cloud, such as gray, varies differently under different lighting conditions. Using gray as the color information to register can cause large errors and even wrong results. To solve this problem, we propose a color point cloud registration algorithm based on hue, which has good robustness at different lighting conditions. We extract the hue component according to the color information of point clouds and make the hue distribution of the tangent plane continuous. The error function consists of color and geometric error of two point clouds under the current transformation. We optimize the error function using the Gauss–Newton method. If the value of the error function is less than the preset threshold or the maximum number of iterations is reached, the current transformation relationship is required. We use RGB-D Scenes V2 dataset to evaluate our algorithm and the results show that the average recall of our algorithms is 8.63% higher than that of some excellent algorithms, and its RMSE of 14.3% is lower than that of the other compared algorithms.

**Keywords:** color point cloud; hue; ICP; registration

## 1. Introduction

With the development of three-dimensional scanning equipment, reconstruction technology has been more and more widely used. For example, dental repairs in the medical field, virtual reconstructions of roles in the film and television industry, and restoration of cultural relics in archaeology are all areas that have found application. Reconstruction is an important method of modeling in which registration technology plays a crucial role. Because the collection field of the three-dimensional scanning device is limited, only part of the whole scene data can be obtained in one operation. In order to gain complete data, collection operations must be performed several times and then spliced together to obtain the whole data. The point cloud is an important form of data, and its registration is the process to estimate the transformation between different point cloud according to the overlap between them, then different data collected at the different positions can be spliced together according to the transformation relationship between the point clouds to form a whole scene data.

One of the most classical methods for this task is the Iterative Closest Point (ICP) algorithm proposed by Besl [1]. ICP uses the nearest point principle to create corresponding point pairs between the target and source point cloud under the current transform, then obtains a new transform by minimizing the mean square error (RMSE) between the corresponding point pairs and updates the current transform. This process is repeated until the best transform is obtained. the ICP algorithm has been widely used in engineering because of its simplicity, efficiency, and accuracy, but the use of coordinate information alone is not enough to constrain the optimization process. For example, when two point clouds with duplicate structures are registered, the alignment effect using ICP algorithm for

point cloud registration is not good, which may cause sliding, resulting in a large error [2,3]. Therefore, in recent years, many extended ICP algorithms [4–7] based on the classical ICP algorithm have been derived.

In recent years, the neural network has made great achievements in image and point cloud processing, and its combination with ICP algorithm is an important trend in point cloud registration. Two studies [8,9] used the neural network method to improve the traditional ICP algorithm by making correspondence through learned features of the point cloud. Although these two algorithms work well, they require a long training time on the dataset. When applying the algorithm, the result will not be good if the test data have large differences compared with training data. Therefore, the improved ICP algorithm based on the neural network has strict limitations on data.

With the development of sensor technology, many industrial 3D scanners are equipped with color cameras, which can acquire the color information of point clouds, which is helpful for registration. A common way to introduce color information to registration is to extend the coordinate with color information and form a four or six-dimensional space, which contains color information [10–13]. However, for the points that have large distances but similar colors, it will sometimes introduce an incorrect correspondence, which will cause the error function to converge in the wrong direction, thus obtaining the incorrect final transformation. The method [14] proposed by Park J et al. solves this problem by establishing a corresponding relationship through the nearest point principle, and like Point-to-Plane ICP, they use the color distribution on the tangent plane to introduce the color information. The error function consists of color and geometric parts, which makes it possible to unify the color and geometric information in registration.

However, the color information used by the algorithm [14] is gray. When the lighting conditions change, the gray of the point cloud would change substantially, and this would cause the error function to converge in the wrong direction, even resulting in the wrong solution. To solve this problem, we present a color point cloud registration algorithm based on hue, which converts the color information of point cloud from gray to hue for registration in order to improve the robustness under varying lighting conditions.

## 2. Color Theory Analysis

RGB color space contains three independent basic colors and is currently the most widely used color space in computer vision. However, it does not reflect human visual perception very well, so it is not efficient when dealing with 'real world' images. For point cloud collected from different locations, it is required that the color is the same, otherwise, the optimization will be led in the wrong direction, producing the wrong results. However, due to the influence of lighting conditions, the color represented by red, green, and blue usually change a lot when the lighting changes. In order to use color information correctly, the difference caused by lighting conditions should be reduced.

HSV color space is based on three components, hue (H), saturation (S), and value (V), and is used to approximate how people perceive and understand colors [15–18]. Hue is represented by an angle between 0 and 360 degrees, and its normalized formula from RGB color space is as follows:

$$H = \begin{cases} 0, & if \quad max(R,G,B) = min(R,G,B) \\ \dfrac{1}{6}\dfrac{G-B}{max(R,G,B)-min(R,G,B)}, & if \quad max(R,G,B) = R \quad and \quad G \geq B \\ \dfrac{1}{6}\dfrac{G-B}{max(R,G,B)-min(R,G,B)}+1, & if \quad max(R,G,B) = R \quad and \quad G < B \\ \dfrac{1}{6}\dfrac{B-R}{max(R,G,B)-min(R,G,B)}+\dfrac{1}{3}, & if \quad max(R,G,B) = G \\ \dfrac{1}{6}\dfrac{R-G}{max(R,G,B)-min(R,G,B)}+\dfrac{2}{3}, & if \quad max(R,G,B) = B \end{cases} \quad (1)$$

It can be seen that for each case, the hue information maps the ratio of the difference between the two smaller components of the RGB, which represents a kind of proportional information. When the ratio of the three color components of the RGB is fixed, the hue information does not change with the intensity of the color component, so it represents the relative brightness of the image. Because the changes caused by lighting conditions affect each component of RGB equally and the ratio of each component's change is fixed, using the relative brightness of RGB to represent color information can reduce the impact of lighting condition, so the hue has better robust than the gray information when representing absolute shading in different lighting conditions. This theory is then proved through experiments.

Figure 1 shows the gray and hue of the same scene under different lighting conditions. The difference map represents how much it would change under different lighting conditions. It is obvious that hue information changes much less than gray, which demonstrates that hue has better lighting robustness than gray. We choose the sum square distance (SSD) of two point clouds to measure the color similarity between them. It can be calculated by the following formula:

$$SSD(P, Q) = \frac{1}{N} \sum_{i=1}^{N} (C(p_i) - C(q_i))^2 \tag{2}$$

where $p_i$ and $q_i$ represent the corresponding point in two point clouds $P$ and $Q$, respectively, and $C(\cdot)$ represents the color information used. Since the point clouds collected are the same scene and only the lighting conditions are different, the color information at the same location should theoretically be the same, so a smaller SSD represents better robustness.

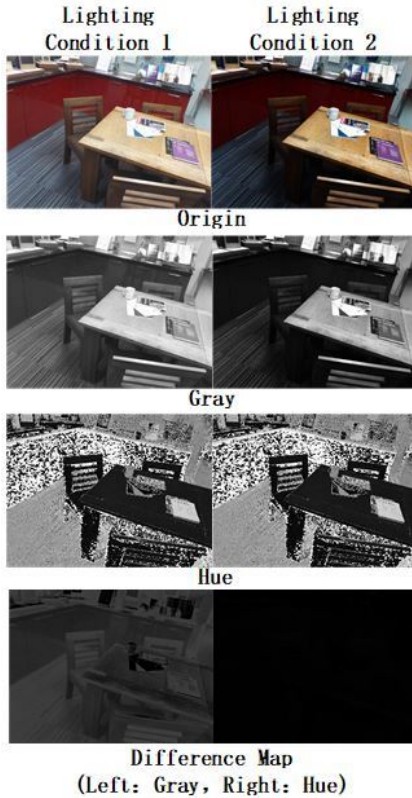

**Figure 1.** The comparison of the same scene at different lighting conditions.

We used the ASTRA Pro RGBD Scanner depth camera to collect the data. The camera position and the layout of the scene are kept unchanged, and different lighting conditions are simulated by changing the exposure time of the camera. First, take a frame as a reference

point cloud, then change the camera exposure time to take multiple pictures under different lighting conditions.

Figure 2 shows the SSD of gray and hue under different lighting conditions. The x-axis of the figure represents the exposure time of the camera, representing different lighting conditions. It can be seen that the SSD of hue is three orders of magnitude lower than gray, which fully demonstrates that the hue is more robust to lighting conditions than gray.

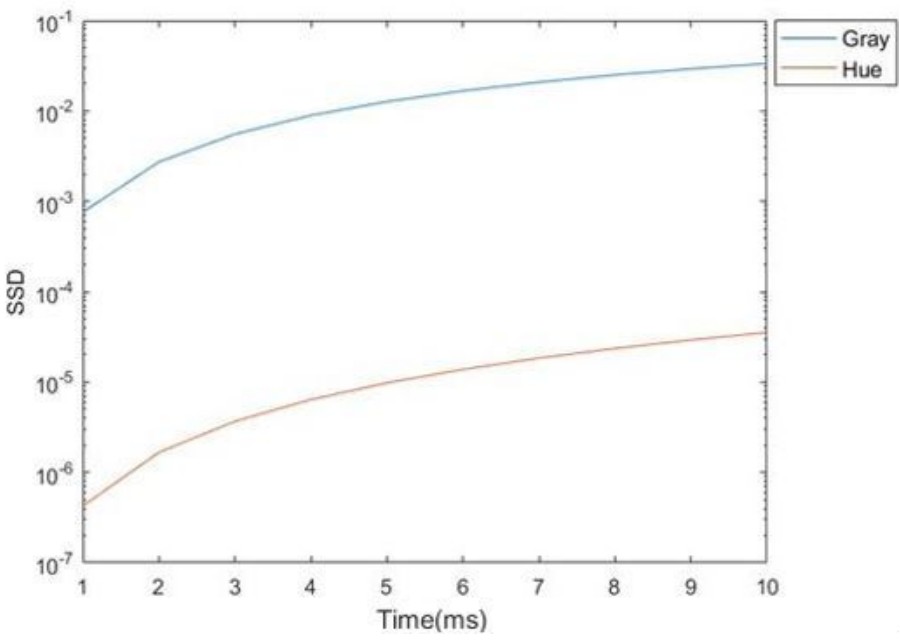

**Figure 2.** The SSD of gray and hue varying with different lighting conditions.

## 3. Algorithm Implementation

We designed a color point cloud registration algorithm that is suitable for different lighting conditions. The algorithm process is shown in Figure 3. The input of the algorithm is two point clouds, the target point cloud P and the point cloud Q. The initial part of the algorithm consists of three steps: color space conversion from RGB to hue for P and Q, color information continuation, and depth information continuation only for point cloud P. The main content of the algorithm is a loop. Firstly, apply the current transformation to Q and get a new point cloud Q′. Then, the corresponding point pairs of P and Q′ are determined using the nearest point principle [7]. Then, calculate the error function in the current iteration and if the error function is smaller than the preset threshold, the iteration stops, and the current transformation is output. Otherwise, the current transformation is updated with gradient descent and the iteration continues.

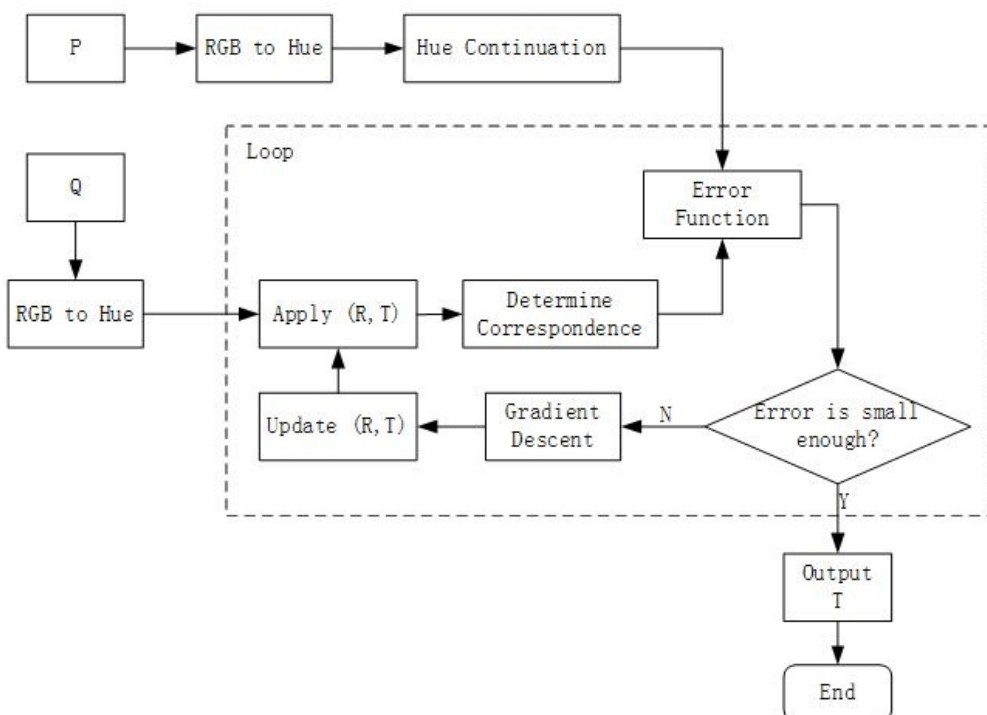

**Figure 3.** The algorithm process.

*3.1. Continuation of Hue Distribution*

$P$ is a point cloud including many points, $p$ is one point of it, and $H(p)$ represents the hue of $p$. When optimizing the error function, the gradient is needed, so we have to convert $H(p)$ into continuous forms from discrete. As for any point $p \in P$, $n_p$ is the normal of $p$, and $\mathcal{N}_p$ is the neighbourhood of $p$. Each point in $\mathcal{N}_p$ has its hue value and the hue distribution of the points is discrete. When $\mathcal{N}_p$ is small enough, we can use the hue distribution on the tangent plane to approximate that in $\mathcal{N}_p$. Let $u$ be one of the vectors on the tangent plane, the origin is point $p$. Because the vector $n_p$ is vertical to the tangent plane, we have $u \cdot n_p = 0$. We assume that $H_p(u)$ is continuous and represents the hue distribution at $p$'s tangent plane. The point $p$ is on the tangent plane and the hue of it is known, and when the radius of $\mathcal{N}_p$ is small, we can normally use first-order Taylor formula at $p$ to approximate $H_p(u)$:

$$H_p(u) \approx H(p) + d_p^T u \tag{3}$$

where $d_p^T$ is the gradient of hue distribution at p and can be estimated through the least square method. As for point $p \in \mathcal{N}_p$, $f(s)$ is the function which project the point $s$ to the tangent plane:

$$f(s) = s - n_p(s - p)^T n_p \tag{4}$$

We can project every point $p'$ in $\mathcal{N}_p$ to the tangent plane of $p$, the projection coordinates are $f(p')$, and we can subtract it with the origin $p$ to transform it to the vector on the tangent plane. In order to approximate the hue distribution of $\mathcal{N}_p$ with that on the tangent plane, we should minimize the hue difference at $p'$ and $f(p')$. The former is known as a constant and the latter can be calculated through Equation (3). $d_p^T$ in the equation determines the continuous distribution but is unknown and we can minimize the hue difference to solve it. The hue difference can be calculated related to $d_p^T$ as following:

$$
\begin{aligned}
L(d_p) &= \sum_{p' \in \mathcal{N}_p} (H_p(f(p') - p) - H(p'))^2 \\
&\approx \sum_{p' \in \mathcal{N}_p} H(p) + d_p^T(p' - p) - H(p'))^2
\end{aligned}
\tag{5}
$$

Because $d_p^T$ is on the tangent plane of $p$, there is a constraint:

$$d_p^t n_p = 0 \tag{6}$$

Ideally, $L(d_p) \approx 0$ and the goal of optimization is to minimize it. This is a linear constrained optimization problem and it can be solved by the Lagrange method.

### 3.2. Error Function

In general, the traditional ICP can get accurate transformation relations on the basis of coarse registration. ICP belongs to the greedy algorithm, and because of the corresponding relationship determined by the nearest point principle, it is easy to fall into the local optimal solution in iteration, which usually occurs in the parts with similar geometric structure. For example, when the source and target point cloud are two planes, there are many solutions if they are on the same plane. Traditional error function only contains the geometric information and may lead to the local optimal solution in the mentioned condition previously.

The error function we used includes not only geometric information but also color information. In the previous chapter, we introduce the robustness of hue and it proves that hue has great robustness about the lighting condition which is usually different. Hence, we use the hue as the color information to constrain the error function.

Let $w \in R^{3 \times 1}$ be the rotation angle of the current transformation and $t \in R^{3 \times 1}$ be the translation vector. The error function is about the transformation $(w, t)$ and the expression of it is as follows:

$$E(w, t) = E_H(w, t) + \sigma E_G(w, t) \tag{7}$$

where $E_H(w, t)$ represents the hue error at the transformation $(w, t)$ while $E_G(w, t)$ is the geometric error, and $\sigma$ is a scalar factor, which is aimed to adjust them to the same order of magnitude to balance their contribution to the optimization. According to the statistics of the point cloud, the range of hue is [0, 1] and that of geometry is [0, 0.03], and adding them directly will result in enhancing the hue error and ignoring the influence of geometric error, considering their range, where $\sigma = 30$ is an appropriate value.

Let $P$ and $Q$ be the target and source point cloud and our goal is to find the transformation that transforms $Q$ to $P$. $\mathcal{K} = (p, q)$ be the correspondence set according to the nearest point principle [1], $p \in P$ and $q \in Q$. Those pairs with the distance beyond the preset threshold are removed. $q'$ is the point $q \in Q$ through the transformation $(\omega, t)$, and their relationship is as follows:

$$q' = s(q, w, t) = R(w)q + t \tag{8}$$

where $R(w) \in R^{3 \times 3}$ is the rotation matrix of $w$, and $s$ represents the transformation with rotation angle $w$ and translation vector $t$. After the transformation, $q'$ is transferred to the target point cloud $P$ from source point cloud $Q$ and can find its correspondence point $p \in P$. Moreover, if the transformation is correct, the target point cloud $P$'s hue at $q'$ would be very similar to that at $p$. For the common condition, $q'$ would not fall at $p$ but in its neighborhood $\mathcal{N}_p$, so we should find the hue information at $q'$ in the neighborhood of $p$. According to Equations (3) and (5), we can approximate the hue distribution in $\mathcal{N}_p$ with that on the tangent plane of $p$, $H_p(u)$, which can be estimated through Equation (4), so we should find the projection of $q'$ at the tangent plane. Let $q''$ be the projection of $q'$ on the tangent plane of $p$:

$$q'' = f(q') \tag{9}$$

The space relationship of point $p, q, q'$ and $q''$ is shown as Figure 4. Point $q$ is in the source point cloud $Q$, after transformation $(w, t)$, and $q'$ is in the target point cloud $P$. Point $q''$ is the projection of $q'$ and is on the tangent plane of $p$. As for every point pair $(p, q)$ in $\mathcal{K}$, the hue error is the difference between the target point cloud's hue at $q''$ and source point cloud's hue at $q$:

$$e_H^{(p,q)} = H_p(q'' - p) - H(q)$$
$$= H_p(f(s(q,w,t)) - p) - H(q) \tag{10}$$

where $H(q)$ is known as the hue information of the source point cloud, and $p$ is the origin of its tangent plane so vector $q'' - p$ is on the tangent plane and, thus, it can be substituted into Equation (3) to obtain the hue information at $q''$ on the tangent plane.

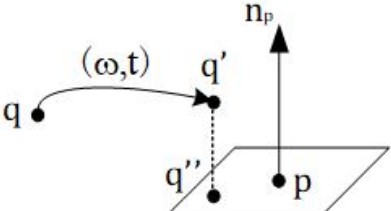

**Figure 4.** The space relationship of the points.

The geometric error is similar to the Point-to-Plane ICP [19], which is the distance from $q'$ to the tangent plane of $p$:

$$e_G^{(p,q)} = (q' - p)n_p$$
$$= (s(q,w,t) - p)n_p \tag{11}$$

In the ideal condition, $q'$ should be very close to $p$, but for most cases, it is impossible, so using the distance from $q'$ to $p$'s tangent plane is suitable to measure the geometric error.

Each pair of points in $\mathcal{K}$ does, above, and considering the continuity of derivation, summing them as a sum of square results, then $E_H$ and $E_G$ are as follows:

$$E_H(w,t) = \sum_{(p,q) \in \mathcal{K}} e_H^{p,q2} \tag{12}$$

$$E_G(w,t) = \sum_{(p,q) \in \mathcal{K}} e_G^{p,q2} \tag{13}$$

According to the Equations (12) and (13), the error function is as follows:

$$E(w,t) = \sum_{(p,q) \in \mathcal{K}} e_H^{p,q2} + \sigma \sum_{(p,q) \in \mathcal{K}} e_G^{p,q2} \tag{14}$$

*3.3. Optimization Method*

The optimization aims to find the optimal solution through minimizing the error function. It is a convex optimization process and we can use Gauss–Newton Method to solve it. The transformation $(w,t)$ consists of 6 independent variables $\xi = (w;t)$, where $w = (\varphi, \theta, \psi)^T$ is the rotation angle of the three axes and $t = (a,b,c)^T$ is the translation vector. The rotation can be divided into three independent rotation around these axes:

$$R(w) = R_x(\varphi)R_y(\theta)R_z(\psi) \tag{15}$$

where $R_x(\varphi)$ represents the rotation of $\varphi$ around the x-axis, and $R_y(\theta)$ and $R_z(\psi)$ are the same operation on the y-axis and z-axis, respectively.

Because the error function consists of two parts, which are hue error and geometric error, the Jacobian matrix can be divided into two parts, $J_{e_H}$ and $J_{e_G}$, which are about the hue and geometry respectively:

$$J_e = \begin{bmatrix} J_{e_H} \\ \sqrt{\sigma}J_{e_G} \end{bmatrix} \tag{16}$$

Let $\xi^k = (\varphi^k; \theta^k; \psi^k; a^k; b^k; c^k)$ be the current transformation after the $k$th iteration, and the Jacobian matrix can be seen as the gradient of error function about the transformation $\xi$, so the Jacobian matrix at $\xi^k$ can be calculated as follows:

$$J_{e_H} = \nabla e_H^{(p,q)}(\xi)|_{\xi=\xi^k} \tag{17}$$

$$J_{e_G} = \nabla e_G^{(p,q)}(\xi)|_{\xi=\xi^k} \tag{18}$$

According to the chain rule, $\nabla e_H^{(p,q)}$ and $\nabla e_G^{(p,q)}$ can be calculated as follows:

$$\begin{aligned}\nabla e_H^{(p,q)}(\xi) &= [\frac{\partial}{\partial \xi_i}(H_p(f(s(q,w,t))-p)-H(q))] \\ &= \nabla H_p(f)J_f(s)J_s(\xi)\end{aligned} \tag{19}$$

$$\begin{aligned}\nabla e_G^{(p,q)}(\xi) &= [\frac{\partial}{\partial \xi_i}(s(q,w,t)-p)^T n_p] \\ &= n_p^T J_s(\xi)\end{aligned} \tag{20}$$

where $\nabla H_p(f) = d_p$ according to Equation (3) and $J_f(s)$ is the Jacobian matrix of $f$ about $s$ while $J_s(\xi)$ is that of $s$ about $\xi$. They can be found by the Equations (3) and (11). The optimization method we used is Gauss–Newton method, and the transformation $\xi$ is updated as follows:

$$\xi^{k+1} = \xi^k - (J_e^T J_e)^{-1}J_e^T e \tag{21}$$

where $e = \begin{bmatrix} e_H \\ \sqrt{\sigma}e_G \end{bmatrix}$. $e_H$ and $e_G$ are the matrix formed through the hue and geometric error according to Equations (10) and (11).

The whole optimization is a process of iteration, and if the number of iterations reaches the maximum or the value of error function is less than the threshold preset, then iteration should be stopped and output the current transformation as the result; otherwise, the iteration will continue until reaching the stopping criteria.

## 4. Experiment

### 4.1. Point Cloud Registration

The experimental environment is Visual Studio 2015, using Open3D point cloud library, the operating system is Windows 10, and computer hardware configuration is Intel Core i9-7980XE CPU, 32GB memory. According to algorithm [14], the experimental parameters are set as follows: the scalar factor $\sigma = 30$, the maximum number of iterations is set to 90, and the threshold of the error function is set to $10^{-6}$.

Figure 5 shows two randomly selected point clouds from the existing dataset [14] and the registration result of the different algorithms. Figure 5a,b are the source and target point cloud, respectively. Figure 5d shows the result through the traditional Point-to-Plane ICP [19]. The result is wrong because the overlap part of two point clouds are not spliced very well. This is because source and target point cloud have too much similar geometric structure such as ground and wall, and they cannot be registered without color information. Moreover, under different collection positions with various lighting conditions, the color information of the overlapping parts of the two point clouds varies greatly, which is obvious in (a) and (b). The error function of [14] using gray as color information will lead to the wrong direction during the iterative optimization process and gain a local optimal result, as shown in Figure 5c. However, as we mentioned previously, the hue is robust to light and has little difference in different lighting conditions. Hence, it would strictly constrain the convergence direction during the optimization process and finally obtain the correct result, as shown in Figure 5g. Figure 5e,f show the results of improved DeepICP [8] and DCP [9] based on the neural network, respectively. We use the pre-trained model in their paper and

apply it to these two point clouds. It can be seen that the results are not appropriate and this is because the deep-learning method relies on the dataset very much and may perform well using one kind of data but not another kind. Their training dataset is CAD models and not scene datasets.

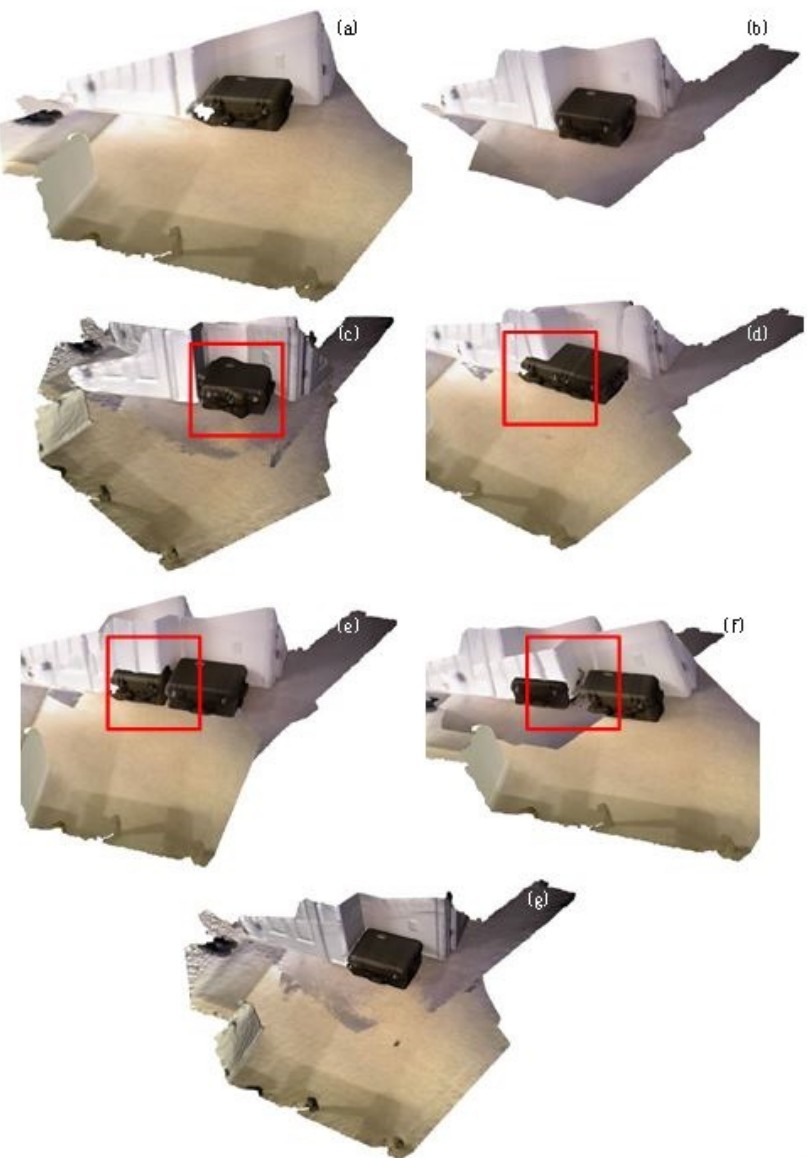

**Figure 5.** The registration results of the two random point clouds from dataset [14]: (**a**) source point cloud and (**b**) target point cloud. Registration results of (**c**) algorithm [14] (RMSE = 0.010607), (**d**) algorithm [20] (RMSE = 0.020071), (**e**) DeepICP (RMSE = 0.031654), (**f**) DCP (RMSE = 0.043145), and (**g**) ours (RMSE = 0.006683).

The common evaluation criterion for point cloud registration is Root Mean Square Error (RMSE). Given two scene fragments $(P_i, P_j)$, their correspondence is $\mathcal{K}_{ij}$, then RMSE under the rigid transformation $\xi_{ij} = (w_{ij}, t_{ij})$ is as follows:

$$RMSE(\mathcal{K}_{ij}, \xi_{ij}) = \sqrt{\frac{1}{\mathcal{K}_{ij}} \sum_{(p,q) \in \mathcal{K}_{ij}} \|R(w_{ij})p + t_{ij} - q\|^2} \tag{22}$$

It can be seen that RMSE represents the difference between corresponding points of the two point clouds, and the smaller it is, the higher the registration accuracy.

Given two scene fragments, if the resulting rigid transformation $\xi_{ij} = (w_{ij}, t_{ij})$ and the true rigid transformation $\xi_{ij}^*$ is close enough, the rigid transformation is correct. $\mathcal{K}_{ij}^*$ is the correspondence under the true transformation $\xi_{ij}^*$. If the RMSE of $\mathcal{K}_{ij}^*$ under the transformation $\xi_{ij}$ is smaller than the given threshold $\tau$, $\xi_{ij}$ is correct:

$$RMSE(\mathcal{K}_{ij}^*, \xi_{ij}) < \tau \tag{23}$$

The dataset used is RGB-D ScenesV2 [13], and each frame contains depth information, color information, and the ground truth transformation matrix of the current frame. Five scenes are selected in this experiment. All of them are point clouds obtained by the RGBD camera scanning indoor scenes. Each frame has slightly different lighting conditions at different viewpoints. For each of the five selected scenarios, 60 frames are selected as test data equidistantly.

We choose Point-to-Plane ICP, colored ICP, and learning method DeepICP and DCP with pre-trained parameters as the benchmark. Because of the sensitivity of the initial transformation, we use FPFH and 5-Point RANSAC to coarse register the point cloud, then use the algorithm above to them.

The source and target point cloud to be registered we selected in the experiment should be at least 30% overlapped thus there are enough correspondence points to calculate the transformation. To evaluate the performance on the dataset, we use Recall in addition to RMSE. The Recall of the registration is the ratio of the successful registration, which means that the transformation calculated should satisfy the Equation (23). Tables 1 and 2 show the recall of the algorithms when threshold $\tau = 0.5$ and the RMSE, respectively. It can be seen that [14] uses gray as color information which did not improve the accuracy of registration because of the influence of different lighting conditions. Scene 01 and Scene04 are even worse than Point-to-Plane ICP. Our algorithm converts color information from gray to hue, and it solves the impact of different light conditions to a certain part. From the data in Table 1, it can be seen that the recall of our algorithm is significantly improved. Table 2 shows that for each scenario, the RMSE of this algorithm is lower than that of algorithm [14], which fully demonstrates the higher accuracy of this algorithm. We also compare two deep-learning-based algorithm. We use the pre-trained model from their paper and test them on our dataset, and the results show that they do not perform well on our dataset, which is very different from their training dataset.

**Table 1.** Recall ($\tau = 0.5$).

| Scene | ICP(Point-to-Plane) | Colored ICP | DeepICP | DCP | Ours |
|-------|---------------------|-------------|---------|-----|------|
| Scene 01 | 0.30169 | 0.2565 | 0.22677 | 0.25655 | 0.31243 |
| Scene 02 | 0.46729 | 0.49933 | 0.39056 | 0.45560 | 0.57276 |
| Scene 03 | 0.40828 | 0.43826 | 0.43055 | 0.39081 | 0.43183 |
| Scene 04 | 0.72208 | 0.64827 | 0.59185 | 0.57669 | 0.69705 |
| Scene 05 | 0.39475 | 0.44788 | 0.46891 | 0.48991 | 0.47814 |
| Average | 0.45882 | 0.458047 | 0.42173 | 0.43391 | 0.498444 |

**Table 2.** RMSE.

| Scene | ICP(Point-to-Plane) | Colored ICP | DeepICP | DCP | Ours |
|-------|---------------------|-------------|---------|-----|------|
| Scene 01 | 0.011659 | 0.010644 | 0.011566 | 0.0133556 | 0.0083734 |
| Scene 02 | 0.009786 | 0.008685 | 0.010001 | 0.012001 | 0.0078229 |
| Scene 03 | 0.010901 | 0.009949 | 0.013001 | 0.009551 | 0.0090334 |
| Scene 04 | 0.009115 | 0.009015 | 0.009511 | 0.010331 | 0.0072203 |
| Scene 05 | 0.013106 | 0.011067 | 0.015600 | 0.011331 | 0.009821 |
| Average | 0.011198 | 0.009872 | 0.011936 | 0.013140 | 0.008454 |

### 4.2. Scene Reconstruction

In order to verify the applicability of the algorithm, this paper also uses real-world scenarios to do experiments. The experimental scene is an office, using an RGBD camera to collect point clouds with color information. The RGBD camera we used is ASTRA Pro RGBD Scanner. Its shooting distance is 0.6 m to 2 m, within which the accuracy is 1 mm. We collect the scene information continuously and make sure the consecutive frames of point cloud have enough overlap (more than 70%) so that we can estimate their transformation. The software part of the camera uses the manufacturer-adapted SDK to acquire depth and color images then convert them to color point clouds.

The process of scene reconstruction is shown in Figure 6. In the beginning, we estimate the transformation of the first tow point cloud and then transfer them to the same coordinate system and splice them together as a new 'big' point cloud. Then, we use the algorithm to estimate its transformation and the next frame, then splice them together and form a bigger point cloud. This operation is performed until all frames are spliced together and we can obtain the whole reconstructed scene.

The results of the reconstruction through our algorithm is shown in Figure 7. The results show that the algorithm works well on the real scene data and our algorithm can be applied to the splicing of three-dimensional point clouds of the real scene data.

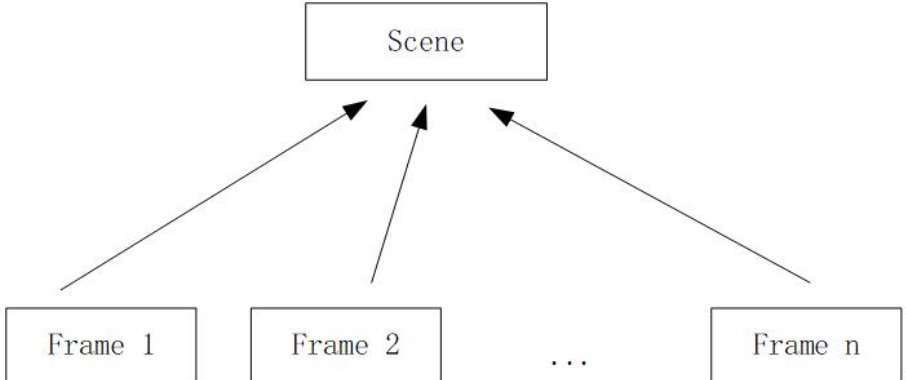

**Figure 6.** The process of reconstruction.

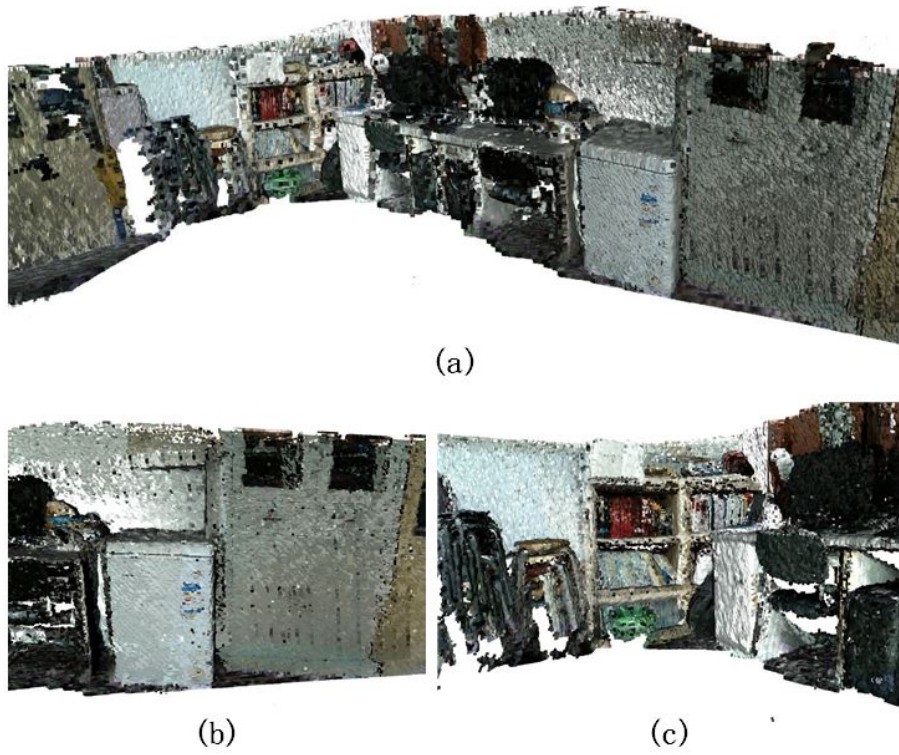

**Figure 7.** The results of reconstruction of the office. (**a**) The whole scene of the office. (**b**,**c**) Different parts of the office.

## 5. Conclusions

A color point cloud registration algorithm based on hue is presented to solve the problem of different lighting conditions. Our algorithm uses hue as the color information to form the error function to obtain the optimal solution. Experiments show that using hue instead of gray can effectively improve the registration success rate and is more robust to light conditions. Using the dataset, the recall of registration is improved by 8.63% compared with the current state-of-art algorithm, and the RMSE of registration is reduced by 14.3%. From the experimental results, the algorithm has good adaptability to light conditions and is more applicable to the actual situation. In addition, real-world scene scans are performed using existing laboratory equipment, and these point clouds are used to further verify the feasibility of the algorithm. At present, because the algorithm searches for corresponding pairs by nearest point matching, the noise of three-dimensional point cloud may lead to inaccurate correspondence, so the algorithm has poor robustness to noise. In the future, our work will focus on this, increasing the robustness of the algorithm to the noise of point clouds.

**Author Contributions:** Conceptualization, S.R.; methodology, S.R. and Y.W.; software, S.R.; validation, S.R., X.C. and H.C.; data curation, S.R., H.L. (Haitao Liang) and H.L. (Haotian Li); writing—Original draft preparation, S.R. and Y.W.; writing—Review and editing, X.C., S.R. and H.C.; supervision, X.C.; project administration, X.C. and Y.W. All authors have read and agreed to the published version of the manuscript.

**Funding:** This research was funded by the National Major Project of Scientific and Technical Supporting Programs of China during the 13th Five-year Plan Period, grant number 2017YFC0109702, 2017YFC0109901 and 2018YFC0116202.

**Institutional Review Board Statement:** Not applicable.

**Conflicts of Interest:** The authors declare no conflict of interest.

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
