# Peer review of "Color Point Cloud Registration Algorithm Based on Hue"

_applsci, doi:10.3390/app11125431_

Round 1

Reviewer 1 Report

The research is on an interesting topic - Color point cloud registration algorithm based on Hue. I believe that it will be useful to researchers and practitioners who are interested in "color point cloud registration algorithm which converts the color information of point cloud from gray to hue for registration, in order to improve the robustness of lighting conditions". The algorithm proposed by the authors is well described, including mathematical formulas, which are explained in detail. The results of the study are also explained in detail.

I have some recommendations for the authors:

  • - to remove the yellow coloring of the texts in the final version of the paper;
  • to expand the theoretical part of the research by adding more citations of authors working in this field;
  • - to review the English grammar before sending the final version of the paper. 

Reviewer 2 Report

The article is very interesting and beneficial for the area of registration of point clouds. I have only a few comments:

p. 4, Fig. 2. - If the X-axis shows exposure time, shoudn't the units be in seconds or miliseconds? What does "um" mean?
p. 11, r. 187 - 70% overlap or 70 frames?
p. 11, Fig. 6. - "Frme 1" should be "Frame 1"
p. 12, Fig. 7. - Whould it be possible to enhance the resolution of the image?
p. 12 - Conclusion - based on what results were the values 4.03 % and 14.3% computed?

Author Response

This manuscript is a resubmission of an earlier submission. The following is a list of the peer review reports and author responses from that submission.

Round 1

Reviewer 1 Report

line 67 empty reference

eq 2 the left hand is missing the arguments

I do not understand fig 2, what is the x-axis?

line 113, the normal is the normal to the surface at point p? please check notation, it is changing each line

line 116, I do not understand what is the hue distribution, is it some probability distribution?what and where is defined the tnagent plane?

line 120, notation is incoherent

line 124 refers to a "previous chapter", looks like some kind of republication of a thesis or similar

It is impossible to decipher the remainder of the section

table 1, I can not find any definition of recall, which is often a performance measure of classification systems, here there is no classification

Reviewer 2 Report

The manuscript presents an integration of the ICP method for the registration of point clouds, with the use of hue information.

The methodology is explained and a test is presented.

The results reported by the authors show a small improvement over the most used methods.

Remarks:

  • The Point Cloud Registration experiment leads to very strange results. In the example, there are al least three planes to use as constraints for the registration, along with several detectable points. Authors should explain how they applied the ICP method. It is known that a selection of vertex pairs must be performed as first step. What point pairs did the authors select?

  • The references are, in most cases, dated. A list of recent papers is as follows:

Hao Zhu , Bin Guo, Ke Zou, Yongfu Li, Ka-Veng Yuen, Lyudmila Mihaylova and Henry Leung. A Review of Point Set Registration: From Pairwise Registration to Groupwise Registration – Sensors - 2019, 19, 1191; doi:10.3390/s19051191

Xiaoshui Huang[1], Guofeng Mei[2], Jian Zhang[2], Rana Abbas[1]. A comprehensive survey on point cloud registration - arXiv - CS - Computer Vision and Pattern Recognition Pub Date : 2021-03-03 , DOI: arxiv-2103.02690

 Miola, G.A., dos Santos, D.R. A framework for registration of multiple point clouds derived from a static terrestrial laser scanner system. Appl Geomat 12, 409–425 (2020). https://doi.org/10.1007/s12518-020-00308-5.

Zan Gojcic, Caifa Zhou, Jan D. Wegner, Leonidas J. Guibas, Tolga Birdal. Learning multiview 3D point cloud registration. Proceedings of the IEEE/CVF Conference on Computer Vision and Pattern Recognition (CVPR), 2020, pp. 1759-1769.

  • The scene reconstruction shows poor quality and is not compared to other solutions.